# Fluorescence Enhancement by Calixarene Supramolecular Aggregate

**DOI:** 10.3390/molecules25245912

**Published:** 2020-12-14

**Authors:** Xin-Yue Hu, Yu-Ying Wang, Hua-Bin Li, Dong-Sheng Guo

**Affiliations:** Key Laboratory of Functional Polymer Materials (Ministry of Education), State Key Laboratory of Elemento-Organic Chemistry, College of Chemistry, Nankai University, Tianjin 300071, China; huxinyue@nankai.edu.cn (X.-Y.H.); kindwyy@mail.nankai.edu.cn (Y.-Y.W.)

**Keywords:** supramolecular assembly, calixarene, fluorescence enhancement

## Abstract

We herein constructed supramolecular assemblies from guanidinocalixarenes and sulfonatocalixarenes by exploiting multiple salt bridge interactions. They encapsulate six different kinds of fluorescent dyes (both cationic and anionic), leading to a fluorescence enhancement that could not be achieved by either single calixarene. As such, this study advances the research on high-performance fluorophores.

## 1. Introduction

The modulation of the photophysical properties of organic fluorophores by supramolecular encapsulation in synthetic macrocyclic hosts attracts much interest and has various applications in sensing, bioimaging, dye laser, and photoluminescent materials, among others [1,2,3,4,5]. The macrocycles crown ether, cyclodextrin, calixarene, cucurbituril, and pillararene are commonly used to investigate the complexation of organic dyes in aqueous media [6]. The host–guest complexation of macrocycles and dyes prompts emission changes, including spectral shift [7,8], fluorescence quenching or enhancement [6,9], and increases in the photochemical stability [8,10]. The effects of inclusion complexation are generally understood in terms of changes in the internal conversion process, related to the relocation of the fluorophore to the hydrophobic environment of the host cavity, prevention of chromophore aggregation, or geometrical confinement of the chromophore within the host [11,12].

The restriction of dye rotation by macrocycles is a representative strategy to enhance the fluorescence of dyes [7,13,14,15,16]. However, in some cases, this approach is hampered by the mismatched size or shape of the macrocycles and dyes. To overcome this limitation, the synergistic restriction of the dye within the macrocycle by other molecules or ions, instead of complexing a single dye in a macrocycle, was explored. For instance, Nau and coworkers enhanced the fluorescence of Brilliant green (BG) by ternary complexation with bovine serum albumin and cucurbit[7]uril (CB[7]), as the fluorescence enhancement by complexation of BG and CB[7] is difficult [17]. Bhasikuttan and coworkers added metal ions to a 2:1 CB[7]:thioflavin T (ThT) complex to strongly increase the fluorescence emission [18], which was not achieved by the ThT and CB[7] complex. Therefore, the synergistic restriction by two or more molecules has emerged as a smart approach to attaining greater fluorescence enhancement in the host–guest assemblies.

Over the past decades, molecular capsule-liked assemblies composed of two or more calixarenes have gained considerable attention [19,20,21]. Owing to the facile modification of the calixarene scaffold, various capsule-like assemblies have been constructed and applied to encapsulation, catalysis, enantioselective recognition, and molecule sensing [21,22,23,24,25,26]. To our knowledge, reports on fluorescence changes resulting from encapsulation into supramolecular calixarene capsule-like assemblies remain elusive, despite the anticipated regulation of the photophysical and photochemical properties by capsulation. In this study, we constructed four calixarene-based assemblies and investigated their ability to enhance the fluorescence of six dyes (Scheme 1), namely 4-[4-(dimethylamino)styryl]-*N*-methylpyridinium iodide (4Asp), 2-[4-(dimethylamino)styryl]-*N*-methylpyridinium iodide (2Asp), 1-anilinonaphthalene-8-sulfonic acid (1,8-ANS), 2-anilinonaphthalene-6-sulfonic acid (2,6-ANS), ThT, and Thiazole orange (TO).

## 2. Results and Discussion

The calixarenes were synthesized by reported procedures (Appendix A) [27,28,29]. Sulfonatocalixarenes SC4A and SC5A, and guanidinocalixarenes GC5A and GC4A, conveniently assemble through the formation of multiple salt bridge interactions between their upper rims, and were combined to construct assembly S4G4 (SC4A + GC4A), S4G5 (SC4A + GC5A), S5G4 (SC5A + GC4A), and S5G5 (SC5A + GC5A) (Scheme 1). Benefiting from the reversibility of non-covalent interactions and the high association constants, simply mixing the solution of different calixarenes generates the assembly spontaneously without further separation and purification. Additionally, most macrocycles exhibit intrinsic substrate selectivity and are rarely compatible with both positive and negative species. Regarding the studied assemblies, guanidinocalixarenes and sulfonatocalixarenes accommodate anionic and cationic guests, respectively. As such, the prepared assemblies may suitably complex both types of species. The strength of the formation of the different assemblies was measured by isothermal titration calorimetry (ITC). In the obtained thermograms, the presence of inflection points at a molar ratio of 1 indicates the formation of 1:1 assembly (Appendix A). The negative Δ*H* and *T*Δ*S* values imply that formation is an enthalpy-driven process (Table 1). In the self-assembly progress, the formation of multiple charge-assisted hydrogen bonds between the anionic sulfonic groups of SC4A/SC5A and the cationic guanidinium groups of GC4A/GC5A, which constitute the upper calixarene rims, dominates the favorable enthalpy change. The unfavorable entropy changes may result from the loss of conformational degrees of freedom upon the assembly formation. Furthermore, the exceptionally high binding constants of approximately 10^7^ M^−1^ indicate a strong affinity between the oppositely charged building blocks.

Next, we studied the typical organic laser dye 4Asp as a putative ideal encapsulated dye (Scheme 1). 4Asp is a common fluorophore that undergoes twisted intramolecular charge transfer (TICT) process [30]. In the local excited state during the photoexcitation, the electron-donating aniline moiety can transfer an electron to the electron-accepting pyridinium unit, and this process is accompanied by single/double-bond twisting [9,31]. Molecules in the TICT state generally favor internal conversion over fluorescent emission [32,33], showing relatively weak fluorescence. Thus, the extremely weak emission of 4Asp in aqueous solution limits its application. First, we investigated whether the encapsulation can enhance the fluorescent intensity of 4Asp. As shown in Figure 1, no fluorescence change was observed by adding GC4A or GC5A into the 4Asp solution. The small fluorescence enhancement observed upon addition of SC4A or SC5A into the 4Asp solution (1.5- to 2.5-fold) suggested that complexation with a calixarene did not effectively restrict the twisting of 4Asp. However, a dramatic fluorescence enhancement (10- to 40-fold) was observed upon the addition of the four capsules into the 4Asp solution, indicating that encapsulation effectively suppressed the TICT. The larger complexation space available in the assemblies and the cooperative interactions of both calixarenes restrict the rotational freedom of 4Asp, and thus hamper the formation of the TICT state. Moreover, the complexation of 4Asp by S4G4 was associated with the largest fluorescence enhancement, presumably because S4G4 had the most suitable size among the tested aggregates. The relative fluorescence quantum yield is 1.49% (Appendix A), nearly seven-fold that of pure 4Asp under the same condition (approximately 0.2%) [34]. Molecular modeling studies (CHARMM36) indicate that SC4A and GC4A can form the heterotopic 1:1 capsule-like structure in a “head-to-head” mode. 4Asp can enter the cavity of the capsule and form a sandwich-liked assembly structure (Figure 2).

In order to prove the synergistic rotational restriction of 4Asp by GC4A and SC4A, we further investigated the encapsulation of 4Asp by S4G4. The solubility of the capsule prevents the direct determination of the 4Asp/S4G4 association constant (*K*_4_, Scheme 2) by fluorescence or ITC titration. Instead, its value was obtained from a thermodynamic cycle. As shown in Scheme 2, the interaction between 4Asp and S4G4 to form 4Asp@S4G4 can proceed via two pathways. In the first one, the complexation of 4Asp with SC4A (4Asp@SC4A, *K*_1_) and subsequently with GC4A delivers 4Asp@S4G4 (*K*_2_), whereas the second pathway involves the initial formation of S4G4 from SC4A and GC4A (*K*_3_), followed by the encapsulation of 4Asp (*K*_4_). As the starting materials and products are the same, the following equation is true:(1)$$K_{4}\;=\;\left(K_{1}\;\times \;K_{2}\right)\;/\;K_{3}$$

*K*_1_ ((1.8 ± 0.1) × 10^4^ M^−1^), *K*_2_ ((1.1 ± 0.3) × 10^8^ M^−1^) and *K*_3_ ((8.0 ± 0.2) × 10^7^ M^−1^) were obtained from ITC (Scheme 2, Appendix A). As such, *K*_4_ was calculated as 2.4 × 10^4^ M^−1^. The enthalpy and entropy changes can also be assessed by the thermodynamic cycle (Scheme 2). As *K*_4_ is higher than *K*_1_, the complex of S4G4 and 4Asp has a slightly more positive cooperativity than that of SC4A and 4Asp. The more favorable entropy change (*T*Δ*S*° = 6.7 kcal/mol) of the earlier process likely leads to a more extensive desolvation effect. The presence of both calixarenes reinforces the dye complex and enables additional favorable hydrophobic and electrostatic interactions, which mutually enhance the binding constant and further restrict the dye in the cavity.

Inspired by these findings, we evaluated the potential of the calixarene capsule-like aggregates as general platforms for fluorescence enhancement by investigating their interaction with the cationic dyes 2Asp, ThT, and TO, and the anionic dyes 2,6-TNS and 1,8-ANS (Scheme 1). A quantitative comparison of the fluorescence intensity before and after the addition of the aggregates is presented in Figure 3. 2,6-TNS exhibits extremely weak fluorescence with the emission maximum at 495 nm in aqueous solution (Appendix A). However, the addition of GC4A led to a seven-fold fluorescence enhancement, which sharply increased to a 79-fold enhancement upon titration with S4G4. The complexation-induced fluorescence enhancement of 1,8-ANS, with an emission maximum at 480 nm, is similar to that of 2,6-TNS. While its fluorescence was slightly enhanced upon complexation with GC4A, a 138-fold intensity increase was observed in the presence of S4G4 (Appendix A). Further, 2Asp, an analogue of 4Asp, is a styryl pyridinium dye with an electron donor-acceptor architecture. Its encapsulation by S4G4 is similar to that of 4Asp (Appendix A), and led to a 50-fold increase in the fluorescence intensity. As for ThT, a fluorescent probe for detecting protein folding [35], a TICT process occurs in the excited singlet state. The resulting transition from the fluorescent local excited state to the nonfluorescent TICT state is responsible for the significant quenching of the ThT fluorescence in low-viscosity solvents, such as water. In viscous solvents or in rigid microenvironments, steric hindrance prevents internal rotation, leading to the suppression of the TICT quenching process and thus a higher fluorescence quantum yield. Upon the binding of ThT with S4G4, the slightly enhanced fluorescence (five-fold) indicated that the TICT process can be limited to some extent (Appendix A). Upon the encapsulation of TO by S4G4, a distinct fluorescence peak with a maximum at 565 nm and a 35-fold increase in intensity was observed (Appendix A). We hypothesize that the inclusion of TO into the cavity restricted its rotational freedom, thus inhibiting the non-radiative pathway.

## 3. Materials and Methods

### 3.1. Materials

All reagents and solvents were commercially available and used as received unless purification is otherwise specified. 4Asp was purchased from Heowns (Tianjin, China), 1,8-ANS and 2,6-TNS were purchased from J&K Chemicals (Beijing, China), ThT was purchased from Acros (Tianjin, China), TO was purchased from Sigma-Aldrich (Shanghai, China). 2-[4-(2-hydroxyethyl)piperazin-1-yl]ethanesulfonic acid (HEPES) was purchased from Meryer (Shanghai, China). 2Asp was synthesized according to previous literature [36]. SC4A, SC5A, GC4A and GC5A were synthesized according to previous literature [27,28,29].

### 3.2. Instruments

ITC measurements were examined on Malvern MicroCal PEAQ-ITC (Malvern, Worcestershire, UK). Steady-state fluorescence measurements were recorded in a conventional quartz cuvette (light path 10 mm) on a Cary Eclipse fluorescence spectrophotometer (Agilent Technologies, Inc., Santa Clara, CA, USA) equipped with a Cary single-cuvette peltier.

### 3.3. ITC Experiment

All microcalorimetric titrations between hosts and guests were performed in HEPES buffer (10 mM, pH = 6.0) at atmospheric pressure and 298 K. Nineteen successive injections were made for each titration experiment. A constant volume (2 µL/injection) of guest (or host) solution in a 40 µL syringe was injected into the reaction cell (280 µL) charged with host (or guest) solution in the same HEPES buffer solution.

A control experiment was carried out in each run to determine the dilution heat by injecting a guest (or host) into a pure HEPES buffer containing no host (or guest) molecules. The dilution heat determined in these control experiments was subtracted from the apparent reaction heat measured in the titration experiments to give the net reaction heat. The net reaction heat in each run was analyzed by using the “one set of binding sites” model to simultaneously compute the binding stoichiometry (N), complex-associated constant (*K*_a_), standard molar reaction enthalpy (Δ*H*°), and standard deviation from the titration curve. Generally, the first point of the titration curve was disregarded, as some liquid mixing near the tip of the injection needle is known to occur at the beginning of each ITC run. Knowledge of the complex-associated constant (*K*_a_) and molar reaction enthalpy (Δ*H*°) enabled the calculation of the standard free energy (Δ*G*°) and entropy changes (Δ*S*°) according to
Δ*G*° = −*RT ln K_a_* = Δ*H*° − *T*Δ*S*°
where *R* is the gas constant, and *T* is the absolute temperature.

To check the accuracy of the observed thermodynamic parameters, two independent titration experiments were carried out to afford a self-consistent thermodynamic parameter.

### 3.4. Fluorescence Quantum Yield Measurements

The fluorescence quantum yields were determined using the following formula [37]:$$\varphi_{i}=\;\varphi_{S\;}\times \;\frac{n_{i}^{2}}{n_{S}^{2}}\;\times \;\frac{I_{i}}{I_{S}}\times \frac{1-10^{-A_{S}\left(\lambda_{exc}\right)}}{1-10^{-A_{i}\left(\lambda_{exc}\right)}}$$
where *φ* is fluorescence quantum yield, *A* is the absorbance at the excitation wavelength, *I* the area under the fluorescence spectra, and *n* is the refractive index of the solvent in which the sample was collected. The subscripts “*φ_i_*” and “*φ_S_*” refer to the sample of interest and the standard, respectively. Coumarin 153 in EtOH (*φ* = 0.38) was used as a standard for 4Asp@S4G4. The excitation wavelength used with Coumarin 153 and 4Asp@S4G4 was 423 nm.

### 3.5. Molecular Simulation

The assemblies of S4G4 and 4Asp@S4G4 were solvated in an equilibrated box of water. The overall charge neutrality was achieved by adding Cl^−^ ions to the solution. The systems before production simulations underwent (i) 5000-step minimizations and 100 ps Molecule Dynamics (MD) simulation with host and guest restrained, and (ii) 2 ns of water equilibration without restraint. Then, the conformation was investigated by performing a 100 ns MD simulation for each complex. The scalable program NAMD 2.13 [38] with the CHARMM36 force field [39] and the TIP3P water model [40] were used to perform the MD simulations. Visualization and analysis of all the MD trajectories were carried out with the VMD program [41].

## 4. Conclusions

In conclusion, we constructed supramolecular assemblies from guanidinocalixarenes and sulfonatocalixarenes, investigated their encapsulation behavior toward several dyes and analyzed the complexation-induced photophysical changes. An increase in the fluorescence intensity was achieved for both cationic and anionic dyes, demonstrating the utility of calixarene assemblies as general platforms for fluorescence enhancement. Such enhancement results from the restriction of the rotational and vibrational freedom of TICT dyes upon encapsulation, which disfavors non-radiative pathways. This novel strategy provides a new approach to achieving high-performance fluorophores, and inspires the rational development of abundant high-performance dyes across different fluorophore families, thus enabling potential applications in bioimaging, advanced organic luminescent materials, high-performance supramolecular dye lasers, and dye-sensitized solar cells.

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
