# Peer review of "Fluorescence Enhancement by Calixarene Supramolecular Aggregate"

_molecules, 2020, doi:10.3390/molecules25245912_

Round 1
Reviewer 1 Report
This manuscript describes the formation of supramolecular capsules from guanidinocalixarenes and sulfonatocalixarenes, and their encapsulation behaviour towards several dyes. The changes induced by the complexation were analysed by fluorescence spectroscopy.
This subject is interesting in the Supramolecular Chemistry area, and the work seems to have been carefully done and the manuscript is well presented. However, Authors used only one method (fluorescence spectroscopy) to study the reported structures.
Thus, in my opinion, this manuscript can be accepted for publication in Molecules after major revision. See also specific comments below:
Specific comments:
- Page 2, line 58: It should be: “and S5G5 (SC5A + GC5A) (Scheme 1).”
- Page 4, Figure 1: the same colour should be used for the 4Asp guest in the four spectra.
- Page 6, Materials and Methods: Although the synthesis of the calixarenes SC4A, SC5A, GC4A and GC5A is already described in the literature, it should be reported, at least in the SI. Some characterization of the compounds (as 1H NMR spectra) should also be included to demonstrate their purity.
- Page 8, lines 229-234: references 12 and 13 have the same article title.
Authors do not report the procedure for the capsule’s formation, mainly in the case of a capsule formed from a calix[4]arene with a calix[5]arene. In addition, Authors should use additional techniques, as NMR, to confirm the proposed structures.
Author Response
This manuscript describes the formation of supramolecular capsules from guanidinocalixarenes and sulfonatocalixarenes, and their encapsulation behaviour towards several dyes. The changes induced by the complexation were analysed by fluorescence spectroscopy.
This subject is interesting in the Supramolecular Chemistry area, and the work seems to have been carefully done and the manuscript is well presented. However, Authors used only one method (fluorescence spectroscopy) to study the reported structures.
Thus, in my opinion, this manuscript can be accepted for publication in Molecules after major revision. See also specific comments below:
Response: We highly appreciate the reviewer’s positive and constructive comments.
Specific comments:
-Page 2, line 58: It should be: “and S5G5 (SC5A + GC5A) (Scheme 1).”
Response: We have corrected this in the revised manuscript.
- Page 4, Figure 1: the same colour should be used for the 4Asp guest in the four spectra.
Response: We have changed the color in Figure 1.
- Page 6, Materials and Methods: Although the synthesis of the calixarenes SC4A, SC5A, GC4A and GC5A is already described in the literature, it should be reported, at least in the SI. Some characterization of the compounds (as 1H NMR spectra) should also be included to demonstrate their purity.
Response: We have added the synthesis and the 1H NMR spectra of calixarenes in the revised supporting information.
- Page 8, lines 229-234: references 12 and 13 have the same article title.
Response: We have corrected the article title of reference 13.
Authors do not report the procedure for the capsule’s formation, mainly in the case of a capsule formed from a calix[4]arene with a calix[5]arene. In addition, Authors should use additional techniques, as NMR, to confirm the proposed structures.
Response: We have added the procedure for capsules’ formation “Benefiting from the reversibility of non-covalent interactions and the high association constants, simply mixing the solution of different calixarenes generates the capsules spontaneously without further separation and purification.” in the revised manuscript.
We agree with the reviewer that NMR is a powerful method to confirm the capsule formation. However, because the solubility of capsules in water cannot reach the concentration for NMR measurement (around micromolar concentration), we cannot use NMR to determine the formation of the capsules. We did the molecular modelling of S4G4 capsule and the 4Asp@S4G4, which indicate the formation of the heteroditopic 1:1 capsule. We added “Molecular modeling studies (CHARMM36) confirm that SC4A and GC4A can form the heterotopic 1:1 capsule and 4Asp can enter the cavity of the capsule (Figure S13).” in the revised manuscript and the molecular simulation structure in Figure S13.
Reviewer 2 Report
The topic of this manuscript is rather interesting and within the scope of the Journal and issue. The possibility to tune fluorescence by using supramolecular systems and capsules is certainly a frontier topic whose knowledge would allow to better design novel analytical systems with enhanced sensitivity.
The paper is well written and contains interesting data comprising enhancement of fluorescence and association constants as determined by ITC. The authors determined a stoichiometry for the assemblies studies (1:1) that should however be intended as relative fluorescence. The conclusions drawn indicate the formation of heterotopic capsules but, to my opinion, there are not too many indications that such assemblies are indeed formed. There are in fact several examples in the literature where calixarenes and resorcinarenes interact forming large discrete assemblies with, for example, six macrocycles. Also other type of assemblies could be envisaged (fibers, micelles …) where the stoichiometry could still be 1:1 (approximately). Authors should bring more solid experimental evidences of the 1:1 heteroditopic capsules they are proposing, for example by using Mass Spectrometry studies or DOSY NMR experiments. Doubts about the formation of 1:1capsules also comes from the possibility of such a small assembly to host, in its interior the proposed fluorophores and influence the TICT behavior. Authors should also try to proposed some models of the free capsules and capsules included with their best guests as obtained by Molecular Modelling studies.
Minor comments:
GC5A, GC4A, SC5A and SC4A might be substituted with simpler G5, G4, S5, S4 codes.
The TICT acronym should be reported indicated also in full the first time it is quoted.
Concerning the statement “As K4 is slightly higher than K1, the complex of S4G4 and 4Asp has a more positive cooperativity than that of SC4A and 4Asp.” It would be quite difficult to me to say this is a cooperative effect. Apparently S4G4 is a slightly better host than SC4A for 4Asp.
Author Response
The topic of this manuscript is rather interesting and within the scope of the Journal and issue. The possibility to tune fluorescence by using supramolecular systems and capsules is certainly a frontier topic whose knowledge would allow to better design novel analytical systems with enhanced sensitivity.
The paper is well written and contains interesting data comprising enhancement of fluorescence and association constants as determined by ITC. The authors determined a stoichiometry for the assemblies studies (1:1) that should however be intended as relative fluorescence. The conclusions drawn indicate the formation of heterotopic capsules but, to my opinion, there are not too many indications that such assemblies are indeed formed. There are in fact several examples in the literature where calixarenes and resorcinarenes interact forming large discrete assemblies with, for example, six macrocycles. Also other type of assemblies could be envisaged (fibers, micelles …) where the stoichiometry could still be 1:1 (approximately). Authors should bring more solid experimental evidences of the 1:1 heteroditopic capsules they are proposing, for example by using Mass Spectrometry studies or DOSY NMR experiments. Doubts about the formation of 1:1capsules also comes from the possibility of such a small assembly to host, in its interior the proposed fluorophores and influence the TICT behavior. Authors should also try to proposed some models of the free capsules and capsules included with their best guests as obtained by Molecular Modelling studies.
Response: We agree with the reviewer that it is important to use DOSY or mass spectrometry to determine the formation of the capsules, however, because the solubility of capsules in water cannot reach the concentration for NMR measurement (around micromolar concentration), we cannot use NMR to determine the formation of the capsules. We also tried the ESI-MS, but did not observe the peak of capsule, which may because the capsules are not stable under the condition of mass spectrometry.
We did the molecular modelling of S4G4 capsule and the 4Asp@S4G4, and the results indicate the formation of the heteroditopic 1:1 capsules. We added “Molecular modeling studies (CHARMM36) confirm that SC4A and GC4A can form the heterotopic 1:1 capsule and 4Asp can enter the cavity of the capsule (Figure S13).” in the revised manuscript and the molecular simulation structure in Figure S13 in the revised supporting information. Our result coincide with those previously reported in the literature(J. Am. Chem. Soc. 2002, 124, 6569-6575; J. Am. Chem. Soc. 2003, 125, 9946-9947; J. Colloid Interface Sci. 2012, 370, 19-26; Chem. Commun.,2009, 4191-4193; J. Am. Chem. Soc. 2004, 126,17050-17058.), which oppositely charged calixarenes prefer to form the heteroditopic capsules.
Minor comments:
GC5A, GC4A, SC5A and SC4A might be substituted with simpler G5, G4, S5, S4 codes.
Response: Thanks for the kind comment. We considered that G5, G4 S5 and S4 could not represent calixarene well and we also want to keep the same abbreviation with our previous research, so we keep the abbreviation.
The TICT acronym should be reported indicated also in full the first time it is quoted.
Response: Thanks for the kind comment. We have changed in the revised manuscript.
Concerning the statement “As K4 is slightly higher than K1, the complex of S4G4 and 4Asp has a more positive cooperativity than that of SC4A and 4Asp.” It would be quite difficult to me to say this is a cooperative effect. Apparently S4G4 is a slightly better host than SC4A for 4Asp.
Response: Thanks for the comments, we have changed this expression “As K4 is slightly higher than K1, S4G4 encapsulates 4Asp more strongly than SC4A.” in the revised manuscript.
Reviewer 3 Report
The paper by Guo et al. describe the use of two calixarene molecules (guanidinio and sulfonato) to “build” capsules able to encapsulate a dye. Upon complexation the dye exhibits a fluorescence enhancement. The authors used ITC and molecular fluorescence spectroscopy to support their claims.
The paper could be of interest nevertheless, some major point has to be clarified.
- Why is there no NMR data to assess the capsule formation and the guest encapsulation inside the capsule? In my mind there is no data that unambiguously demonstrate the formation of a capsule. ITC's measurement only indicate an interaction between the 2 molecules (capsule or not)
- Did the author tried an ESI-MS experiment to try to identify capsules? (not mandatory for the revision but it should be great to prove the capsule formation by either NMR or MS)
- Regarding the quantum yield, it is generally not expressed in percentage but rather by a value between 0 and 1. It means that 1.49% is a quantum yield of 0.0015 which is very very low. The “dramatic 138-fold intensity increase” line 126 could be achieved with other fluorescent off-on systeme so in my opinion the word “extraordinary” in the title is exaggerated and the title should be rewritten.
- There is no reference for quantum yield measurement. How was it measured? It seems that coumarin 153in ethanol was used as standard but the standard should have a quantum yield close to the one of the studied compounds. What is the quantum yield of Coumarin 153 in ethanol?
- In the abstract, authors argue that supramolecular assemblies have many practical applications. It is too general and uninformative. They encapsulate a variety of fluorescent dye : I see two class : pyridinium (cationic) compounds and sulfonate (anionic). In the pyridinium class I see only two different type of family. It is not a variety of dye. Where are the neutral guest (anthracene?) ? Why is there no coumarin? Variety of dye is too vague. Extraordinary fluorescence enhancement is exaggerated. The abstract should be rewritten to be more informative.
- Scheme 1: the scheme is not easy to read. It should be an idea to start from the left with both half capsule, then include the guest and finally, on the right close the capsule. The counter ion is not indicated for cationic species. It has to be.
- Line 61-62: description of the synthesis should be placed before the explanation of capsule.
To conclude, the work could be of interest if the author were able to prove the capsule formation and after some work on the text (and title!).
Reviewer 4 Report
The manuscript “Extraordinary Fluorescence Enhancement by 3 Calixarene Capsules” submitted by Prof Guo and coworkers is well-written and concise. They have shown a significant enhancement in fluorescence of common dyes by self-assembling with calixarenes containing charges (+/-) in each unit, thus producing a capsule effect. This idea is not new (see RSC Adv., 2018, 8, 22530-22535, Acc. Chem. Res. 2014, 47, 1925−1934 (from Prof Guo – corresponding author of this paper), but the molecules used here and the significant improvement in fluorescence quantum yield is remarkable, justifying the publication in molecules. I would suggest the authors quantify better this effect/achievement in terms of fluorescence quantum yield and, after that, I recommend it for publication in Molecules.
Author Response
The manuscript “Extraordinary Fluorescence Enhancement by 3 Calixarene Capsules” submitted by Prof Guo and coworkers is well-written and concise. They have shown a significant enhancement in fluorescence of common dyes by self-assembling with calixarenes containing charges (+/-) in each unit, thus producing a capsule effect. This idea is not new (see RSC Adv., 2018, 8, 22530-22535, Acc. Chem. Res. 2014, 47, 1925−1934 (from Prof Guo – corresponding author of this paper), but the molecules used here and the significant improvement in fluorescence quantum yield is remarkable, justifying the publication in molecules. I would suggest the authors quantify better this effect/achievement in terms of fluorescence quantum yield and, after that, I recommend it for publication in Molecules.
Response: Thanks for the kind comments. We have added some quantification of the effect, which have been highlighted, in the revised manuscript.
Round 2
Reviewer 1 Report
- The text added on page 3, lines 81-83, is not exactly the same written in the “Author response to report 1”. Moreover, I think those lines should be added on beginning of page 3, after “Scheme 1”.
- What about DMSO? Have Authors tried the capsules proton NMR spectra in DMSO?
- Concerning “Molecular Modelling Studies”, Figure S13 should be in the main text, together with a description / discussion of the results obtained (section 2. Results and Discussion”). Also the word “confirm” on page 4, line 117 should be replaced by “indicate”.
Author Response
- The text added on page 3, lines 81-83, is not exactly the same written in the “Author response to report 1”. Moreover, I think those lines should be added on beginning of page 3, after “Scheme 1”.
Reply: Thanks for the advise. We have corrected the word and added it after Scheme 1.
- What about DMSO? Have Authors tried the capsules proton NMR spectra in DMSO?
Reply: Thanks for your kind comment. But we think the interactions between two calixarenes in DMSO is different from in water, the formation of capsule proved in DMSO may not be generalized to water, so we did not choose DMSO as the solvent to do the spectra.
- Concerning “Molecular Modelling Studies”, Figure S13 should be in the main text, together with a description / discussion of the results obtained (section 2. Results and Discussion”). Also the word “confirm” on page 4, line 117 should be replaced by “indicate”.
Reply: We added Figure S13 in the main text as Figure 2 and add a discussion which is highlighted in the revised manuscript. And the word “confirm” was replaced as “indicate”.
Reviewer 2 Report
Despite the request of this referee, the authors did not give convincing evidences of the formation of capsules in this case. No NMR data or ESI/Maldi MS were reported in support to the formation of these discrete supramolecular assemblies. It is true, as claimed from the authors, that the systems is quite similar to those use by other authors and quoted in their answer, but is not the same. In these references the authors could indeed collect both MS and NMR evidences of capsule formation. I understand that solubility is an issue but this is not a reason to claim capsules are forming. Also, in comparison to previous studies, in this study the guest is much larger and, as shown by the models prepared in this second version of the paper, (at least 4Asp) does not allow to maintain the supramolecular contacts between sulfonates and guanidinium.
I would therefore suggest to change the conclusions (and title accordingly) and simply report that some supramolecular assemblies are forming, whose nature cannot be proven.
This change is necessary prior to publication.
Author Response
Despite the request of this referee, the authors did not give convincing evidences of the formation of capsules in this case. No NMR data or ESI/Maldi MS were reported in support to the formation of these discrete supramolecular assemblies. It is true, as claimed from the authors, that the systems is quite similar to those use by other authors and quoted in their answer, but is not the same. In these references the authors could indeed collect both MS and NMR evidences of capsule formation. I understand that solubility is an issue but this is not a reason to claim capsules are forming. Also, in comparison to previous studies, in this study the guest is much larger and, as shown by the models prepared in this second version of the paper, (at least 4Asp) does not allow to maintain the supramolecular contacts between sulfonates and guanidinium.
I would therefore suggest to change the conclusions (and title accordingly) and simply report that some supramolecular assemblies are forming, whose nature cannot be proven.
This change is necessary prior to publication.
Reply:Thanks for the kind advice. We think maybe “capsule” is not suitable for the guest-included case, so we used “sandwich-like aggregate” to describe. We also used “supermolecular assembly” or “aggregate” instead capsule to describe the structure in conclusions and title.
Reviewer 3 Report
The manuscrit was greatly improved during the first round of review. nevertheless, some point still need clarifications:
- “In the abstract, authors argue that supramolecular assemblies have many practical applications. It is too general and uninformative”. This remark was not taken into account in the first round of review. Please indicate some practical application in the abstract to encourage reader to read the full article. practical application were only given at the end of the conclusion.
- Line 185 : there is an “i” missing for n²/ns². n is not defined. A reference for the formula should be provided.
- Title should be changed in SI too
- Thank you for adding synthesis protocols and NMR spectra. It is helpful for the reader.
- Molecular modeling: with molecular simulation, I can believe that a capsule is generated (in the case of S4G4 and S5G5). Nevertheless, with the 4Asp dye (fig S13 (b)), it seems that the capsule does not exist anymore. Both end of the fluorophore are capped but guanidino and sulfonate groups are too far to form a capsule. If the “capsule” is not closed, is it still a “capsule”?
- Authors argue that at micromolar concentration, calixarene will precipitate in solution (D2O) and thus, capsule formation can not be monitored by NMR. Author have to try to prove the existence of their capsules in other solvent like DMSO (or DMSO/D2O mixture) where GC are soluble. Such evidence of capsule formation is mandatory to support the conclusion.
To conclude, without evidence of capsule formation, the first sentence of the conclusion is not supported by data.
I can not recommend publication of the manuscript in the present form
Author Response
The manuscrit was greatly improved during the first round of review. nevertheless, some point still need clarifications:
- “In the abstract, authors argue that supramolecular assemblies have many practical applications. It is too general and uninformative”. This remark was not taken into account in the first round of review. Please indicate some practical application in the abstract to encourage reader to read the full article. practical application were only given at the end of the conclusion.
Reply: We have rewrite the abstract.
- Line 185 : there is an “i” missing for n²/ns². n is not defined. A reference for the formula should be provided.
Reply: Thanks for your advice, and we have corrected and highlight the correction in the revised manuscript, and the reference is also provided.
- Title should be changed in SI too
Reply: We have corrected in SI.
- Thank you for adding synthesis protocols and NMR spectra. It is helpful for the reader.
Reply: Thanks for your advise.
- Molecular modeling: with molecular simulation, I can believe that a capsule is generated (in the case of S4G4 and S5G5). Nevertheless, with the 4Asp dye (fig S13 (b)), it seems that the capsule does not exist anymore. Both end of the fluorophore are capped but guanidino and sulfonate groups are too far to form a capsule. If the “capsule” is not closed, is it still a “capsule”?
Reply: Thanks for the kind advice. We think maybe “capsule” is not suitable for the guest-included case, so we used “sandwich-like aggregate” to describe.
- Authors argue that at micromolar concentration, calixarene will precipitate in solution (D2O) and thus, capsule formation can not be monitored by NMR. Author have to try to prove the existence of their capsules in other solvent like DMSO (or DMSO/D2O mixture) where GC are soluble. Such evidence of capsule formation is mandatory to support the conclusion.
Reply: Thanks for your kind comment. But we think the interactions between two calixarenes in DMSO is different from in water, the formation of capsule proved in DMSO may not be generalized to water, so we did not choose DMSO as the solvent to do the spectra. We used “supermolecular assembly” or “aggregate” instead capsule to describe the structure in conclusions and title.
To conclude, without evidence of capsule formation, the first sentence of the conclusion is not supported by data.
I can not recommend publication of the manuscript in the present form
Reviewer 4 Report
The authors have addressed the fluorescence quantum yield measurements adequately. I recommend the publication as it is.
Author Response
The authors have addressed the fluorescence quantum yield measurements adequately. I recommend the publication as it is.
Reply:Thanks for the kind comment.